# COMPLEX LOCOMOTION SKILL LEARNING VIA DIFFERENTIABLE PHYSICS

## ABSTRACT

Differentiable physics enables efficient gradient-based optimizations of neural network (NN) controllers. However, existing work typically only delivers NN controllers with limited capability and generalizability. We present a practical learning framework that outputs unified NN controllers capable of tasks with significantly improved complexity and diversity. To systematically improve training robustness and efficiency, we investigated a suite of improvements over the baseline approach, including periodic activation functions, and tailored loss functions. In addition, we find our adoption of batching and a modified Adam optimizer effective in training complex locomotion tasks. We evaluate our framework on differentiable mass-spring and material point method (MPM) simulations, with challenging locomotion tasks and multiple robot designs. Experiments show that our learning framework, based on differentiable physics, delivers better results than reinforcement learning and converges much faster. We demonstrate that users can interactively control soft robot locomotion and switch among multiple goals with specified velocity, height, and direction instructions using a unified NN controller trained in our system.

## 1 INTRODUCTION

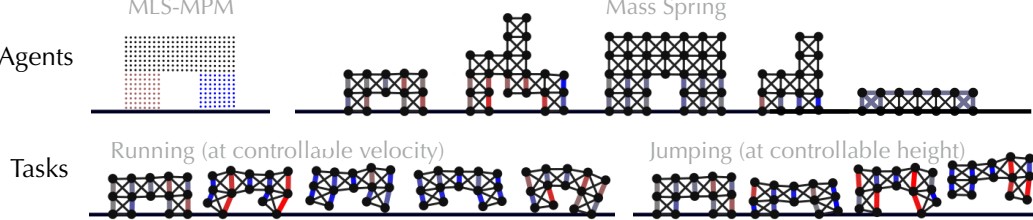

Figure 1: **Our learning system is robust and versatile, supporting various simulation methods, robot designs, and locomotion tasks with continuously controllable target velocity and heights.**

Differentiable physical simulators deliver accurate analytical gradients of physical simulations, opening up a promising stage for efficient neural network (NN) controller training via gradient descent. Existing research demonstrates that on simple tasks, learning systems via differentiable physics can effectively leverage simulation gradient information and converge orders of magnitude faster (see, e.g., de Avila Belbute-Peres et al. (2018); Hu et al. (2019)) than reinforcement learning.

However, the capability of existing differentiable physics based learning systems is relatively limited. Typically, optimized controllers can only achieve relatively simple single-goal tasks (e.g., moving in one specific direction, as in ChainQueen Hu et al. (2019)). Those learned controllers often have difficulty generalizing the task to a perturbed version.

In this work, we propose a learning framework for complex locomotion skill learning via differentiable physics. The complexity of our tasks comes from two aspects: first, the degrees of freedom of our soft agents are significantly higher compared with the rigid-body ones; second, the agents are expected to learn multiple skills at the same time. We systematically propose a suite of enhancements (Fig. 4) over existing training approaches (such as Hu et al. (2020); Huang et al. (2021)) to

significantly improve the efficiency, robustness, and generalizability of learning systems based on differentiable physics.

In addition, we investigated the contributions of each enhancements to training in detail by a series of ablation studies. We also evaluated our framework through comparisons against Proximal Policy Optimization (PPO) Schulman et al. (2017), a state-of-the-art reinforcement learning algorithm. Results show that our system is simple yet effective and has a much-improved convergence rate compared to PPO.

We demonstrate the versatility of our framework via various physical simulation environments (mass-spring systems and the moving least squares material point method, MLS-MPM Hu et al. (2018)), robot designs (structured and irregular), locomotion tasks (moving, jumping, turning around, all at a controllable velocity). To the best of our knowledge, this is the first time when a neural network can be trained via differentiable physics to achieve tasks of such complexity (Fig. 1). Our agent is simultaneously trained with multiple goals, the learned skills can be continuously interpolated. For example, we can control both the velocity and height of an agent while it runs. The trained agent can also be controlled in real-time, enabling direct applications in soft robot control and video games. In summary, our key contributions are listed below:

- To the best of our knowledge, we proposed the first differentiable physics based learning framework for multiple locomotion skills learning on soft agents using a single network.
- We developed an end-to-end differentiable physical simulation environment for deformable robot locomotion skills learning, which supports mass-spring system and material point method as dynamic backends.
- We systematically investigated the key factors contributing to the differentiable physics based learning framework. We believe our investigation can inspire further researches to explore the possibilities of differentiable physics for more complex learning tasks.

## 2 DIFFERENTIABLE SIMULATION ENVIRONMENTS

### 2.1 SIMULATION SETTINGS

Our learning framework supports multiple types of differentiable physically-based deformable body simulators. The simulator can be seen as a black box that outputs the states for the next time step given the actuation signal and states from the current time step. The NN controller takes the information given by the simulator as part of its input features and decides the actuation signal for the simulator in the next time step. With our automatic differentiation system, we can evaluate the gradient information to guide the training of our NN controllers. We use the mass-spring systems and the moving least squares material point method (MLS-MPM) as our simulation environments. The design choices for each environment can be found in the appendix.

### 2.2 AGENT STRUCTURE AND ACTUATION MODE

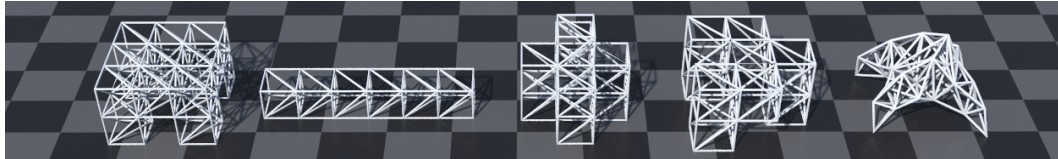

Figure 2: **3D agents collection.** These agents are designed with simple stacked cubes or complex handcrafted meshes.

Our framework supports a variety of differently shaped agents as shown in Fig. 2 and 3. Our actuation signal exerts forces along with the "muscle" directions of the simulated agents. This signal falls within the interval $[-1, +1]$, where its sign determines whether an agent wants to contract (negative sign) or relax (positive sign) a muscle and its absolute value determines the magnitude of the force. In mass-spring systems, the muscle directions are represented by the spring directions, we change the rest-length of springs to generate forces. The actuation signal is used to scale the rest-length up

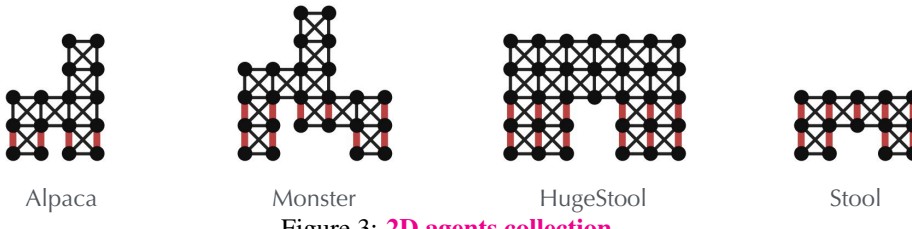

Figure 3: **2D agents collection.**

to some limit (usually at $20\%$). In addition, only activated springs, i.e. actuators (springs marked red or blue shown in Fig. 1), are able to generate forces according to control signals. In material point methods, the muscle directions are always along the vertical axis in material space. In our implementation, we simply modify the Cauchy stress in the vertical direction of material space to apply this force. Again, the force is scaled to a user-defined bound which can be adjusted for different simulations.

## 3 LEARNING FRAMEWORK

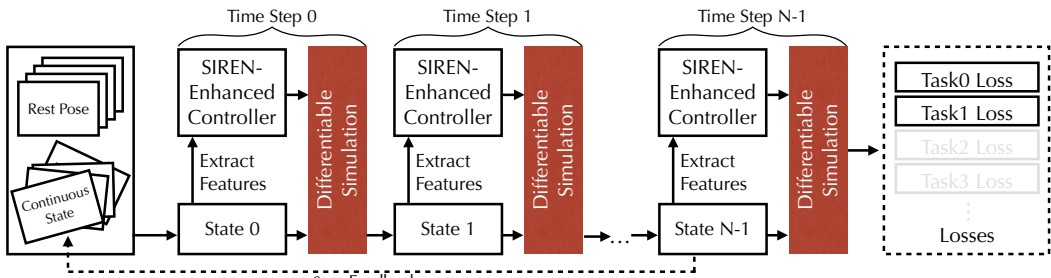

Figure 4: **Framework overview.** Simulation instances are batched and executed in parallel on GPUs. The whole system is end-to-end differentiable, and we use gradient-based algorithms to optimize the neural network controller weights. Each time step involves evaluating a NN controller inspired by SIREN Sitzmann et al. (2020), and a differentiable simulator time integration implemented using DiffTaichi Hu et al. (2020). Simulation states are fed back to the initial state pool to improve the richness of training sets and thereby the robustness of the resulted NN controller. A tailored loss function (as shown in section 3.2) is designed for each task.

In this section, we describe our training framework based on differentiable physics (Fig. 4). In summary, we develop a differentiable simulator with an embedded NN controller. In each time step, the program performs an NN controller inference and a differentiable simulator time integration. After a few hundred time steps, the program back-propagates gradients end-to-end via reverse-mode automatic differentiation, and updates the weights of the neural network using the Adam optimizer Kingma & Ba (2014).

### 3.1 TASK REPRESENTATION

Given an agent $A$, its position and corresponding velocity are denoted as $\mathbf{x} = (x, y, z)$ and $\mathbf{v} = (u, v, w)$. We presents three tasks (running, jumping and crawling) for 2D agents, two (running and rotating) for 3D. In a whole simulation with total steps $\mathcal{T}$, an agent is expected to achieve all goals in a goal sequence $\mathcal{G} = \{\mathbf{G}_1, \mathbf{G}_2...\mathbf{G}_n | n\mathcal{P} = \mathcal{T}\}$, where $\mathcal{P}$ is the time period for an agent to achieve a goal, $n$ is the number of goals . For each goal $\mathbf{G}$, there are multiple tasks in it. Each task is encoded using a target value $g$, which instructs the controller to drive the agent. For example, $\mathbf{G} = \{g_v, g_h\}$ represent performing running and jumping simultaneously, the target velocity and height are $g_v$ and $g_h$. The agent is expected to switch between multiple goals during one simulation.

**Running.** The running task is defined as a velocity tracking problem. Given a time step $t$, the center of mass $\mathbf{c}$ is used to represent the position of the agent, its velocity is defined as the averaged velocity $\tilde{\mathbf{v}}$ over a time period $\mathcal{P}_r$, $(\mathcal{P}_r < \mathcal{P})$. Then the agent is expected to achieve a velocity $\tilde{\mathbf{v}}$ close to the target velocity $g_v$.

$$\tilde{\mathbf{v}}(t) = \frac{1}{\mathcal{P}_r} \left( \mathbf{c}(t) - \mathbf{c}(t - \mathcal{P}_r) \right) \tag{1}$$

**Jumping.** The jumping task is defined as reaching a given height $g_h$ in a time period $\mathcal{P}$. Due to the gravity, it is not possible for a running robot to stay away from the ground. Therefore, the jumping height $\tilde{h}_h(t)$ of an agent is defined by the maximum vertical position of its lowest point in a certain period $\mathcal{P}$. To be more formally, given $\mathbf{S}$ a set of nodes (or points) that constitute the agent, $h$ is a function that can extract the vertical position given a node (or point), the jumping height over the $n$-th period is defined as

$$\tilde{h}_h(n) = \max_{t \in [0, \mathcal{P}]} \min_{s \in \mathbf{S}} h(s, t + n\mathcal{P}) \tag{2}$$

where $n$ is the index of the period in a whole simulation.

**Crawling.** The crawling task is defined as lowering the highest point as much as possible. The target of crawling $g_c$ is defined as an indicator function. The agent is controlled to crawl if $g_c = 1$ otherwise if $g_c = 0$. Since crawling is a status that needs to be maintained, the crawling height is defined as the vertical position of its highest point at each time step.

$$\tilde{h}_c(t) = \max_{s \in \mathbf{S}} h(s, t) \tag{3}$$

## 3.2 Loss functions

Consider a locomotion task where an agent is instructed to move at a specified velocity, we think a proper loss function should satisfy the following requirements:

1. **Periodicity.** The agent is expected to move periodically following a predefined cyclic activation signal. Therefore, the loss function should encourage periodic motions.

2. **Delayed evaluation.** Due to inertia, it takes time for the agent to start running or adjust running velocity. Therefore, an ideal loss function should take this delay into consideration.

3. **Fluctuation tolerance.** During a single running cycle, requiring the center of mass to move at a *constant* velocity at each time step may induce highly fluctuated losses. To avoid that, our loss function should be smooth during one motion cycle.

**Task loss.** For each task, we defined a tailored loss shown in equation 4 according to the requirements above. The loss for running $\mathcal{L}_v$ is defined as the accumulation of the difference between the target and the agent's velocity from start to current time step. For jumping, we define a sparse loss $\mathcal{L}_h$, which only evaluate once for each period. The crawling loss $\mathcal{L}_c$ is defined as the accumulation of the crawling height if the loss is applicable.

$$\mathcal{L} = \lambda_v \underbrace{\sum_{n \in \mathcal{T}} \sum_{t=\mathcal{P}_r}^{\mathcal{P}} (\tilde{\mathbf{v}}(t) - g_v(n))^2}_{\mathcal{L}_v} + \lambda_h \underbrace{\sum_{n \in \mathcal{T}} (\tilde{h}_h(n) - g_h(n))^2}_{\mathcal{L}_h} + \lambda_c \underbrace{\sum_{n \in \mathcal{T}} \sum_{t=0}^{\mathcal{P}} g_c(n) \tilde{h}_c(t + n\mathcal{P})}_{\mathcal{L}_c} \tag{4}$$

The $\lambda_v, \lambda_h, \lambda_c$ are weights for losses of dfferent tasks. Our loss function accumulates the contributions of all steps in a "sliding window". This prevents the loss function from being too small as well as the vanishing gradient problem. We find that having an accumulated loss evaluated at every step works much better when we want to control the speed of the agent explicitly.

**Regularization.** Since the agents may keep shaking when no target velocity or height are given, we introduce an actuation loss as a regularization term. The intuition is to penalize comparatively large actuations when small goals value is given:

$$\mathcal{L}_a = \sum_{n \in \mathcal{T}} \sum_{t \in [0, \mathcal{P}]} (\mathcal{A}(t) - \mu ||g_v(n)||)^2 \tag{5}$$

where $\mathcal{A}(t)$ is the actuation output of NN, $\mu$ is a normalizer. The actuation loss balances the importance of input channels of the NN, i.e., different input channels produce similar contributions to the first layer of the trained NN.

### 3.3 NETWORK ARCHITECTURE

We use two fully connected (FC) layers with sine activation function as the neural network. The network input vector consists of three parts:

1. **Periodic control signal** of the same period with different phases, to encourage periodic actuation. In this work, we use sin waves as the periodic control signal.

2. **State feature vector** extracted from the current state of the agent. Using the positions and velocities of all the vertices of the simulated object as the input feature is not a good idea since these quantities are neither translation-invariant nor rotation-invariant. We use a "centralized pose" to remove global translation and rotation information. To be more specific, we subtract the position of the center of mass (CoM) in both 2D and 3D cases to remove the global translation. In 3D cases, we also compute the rotation around the vertical axis of the agent to remove its global rotation.

3. **Targets** that encode the task information such as running velocity, jumping height, and orientation for 3D agents. Different tasks use different channels to represents their target values. During the training, we assign random values to these targets. When validating the model, we always test the agent on a fixed set of target values. In interactive settings, we assign various target values to control the agent's motion. To amplify the importance of the targets, we duplicate these channels multiple times, try to make them comparably important among the other input features.

### 3.4 TRAINING

In the training process, we run one simulation in one training iteration. The duration of the simulation is divided into several periods $\mathcal{P}$ as mentioned in the section 3.1. For each $\mathcal{P}$, a goal $\mathbf{G}$ contains several tasks is assigned. For example, $\mathbf{G} = \{g_v, g_h\}$ represents that the agent is expected to perform both running and jumping in this period. The goal and its target values are generated randomly in a uniform distribution. They are kept fixed inside one period $\mathcal{P}$, but varied between different periods.

## 4 EXPERIMENTS AND ANALYSIS

In this section, we systematically investigate the key factors that contribute to the training, evaluate our framework through ablation studies and comparisons against PPO Schulman et al. (2017), a state-of-the-art reinforcement learning algorithm. All experiments in the ablation studies are performed at least three times. The results of ablation studies are summarized in Table 1 and Fig. 5.

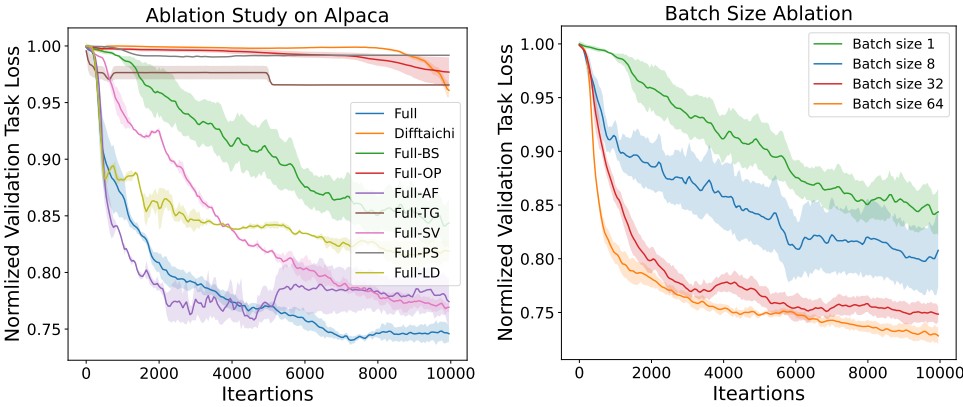

Figure 5: **Summary of the ablation study.** Here we show the summary of ablation study on agent *Alpaca*. Each iteration indicates one training iteration, which is composed of 1000 steps of physical simulation and one step of update on weights of the controller. The result show that the *Full* method achieves the best performance. For more results of ablation studies on other agents, please check the appendix.

| Name | Training Setting | | | | | | | Norm. Valid. Loss (Alpaca) | | |
|---|---|---|---|---|---|---|---|---|---|---|
| | OP | AF | BS | PS | SV | TG | LD | Run. | Jump. | Task |
| Full | Adam | sin | 32 | ✓ | ✓ | ✓ | ✓ | 0.58±0.01 | 0.79±0.01 | 0.74±0.01 |
| Difftaichi* | SGD | tanh | 1 | ✓ | ✓ | ✓ | ✓ | 0.98±0.00 | 0.95±0.01 | 0.96±0.01 |
| Full-BS | Adam | sin | 1 | ✓ | ✓ | ✓ | ✓ | 0.82±0.02 | 0.85±0.01 | 0.84±0.02 |
| Full-OP | SGD | sin | 32 | ✓ | ✓ | ✓ | ✓ | 0.97±0.01 | 0.98±0.01 | 0.98±0.01 |
| Full-AF | Adam | tanh | 32 | ✓ | ✓ | ✓ | ✓ | 0.67±0.02 | 0.81±0.01 | 0.77±0.02 |
| Full-PS | Adam | sin | 32 | ✗ | ✓ | ✓ | ✓ | 0.97±0.00 | 0.99±0.00 | 0.99±0.00 |
| Full-SV | Adam | sin | 32 | ✓ | ✗ | ✓ | ✓ | 0.54±0.01 | 0.84±0.01 | 0.77±0.01 |
| Full-TG | Adam | sin | 32 | ✓ | ✓ | ✗ | ✓ | 0.99±0.00 | 0.95±0.00 | 0.97±0.00 |
| Full-LD | Adam | sin | 32 | ✓ | ✓ | ✓ | ✗ | 0.99±0.01 | 0.74±0.01 | 0.82±0.01 |

Table 1: **Ablation summary on agent *Alpaca*.** In this table, we show the *Task*, *Run* and *Jump* validation loss of proposed method and its ablated versions. *Full* represents the proposed method. The abbreviations under training setting represent name of components to remove. To be more specific, OP: Optimizer, AF: Activation Function, BS: Batch Size, PS: Periodic Signal, SV: State Vector, TV: Targets, LD: Loss Design. *Difftaichi is an implementation of Hu et al. (2020), enhanced with our loss design. It can be observed that the *Full* method achieves the best performance on *Task* loss among all models.

## 4.1 VALIDATION METRIC

In validation, we test all the trained agents on a fixed set of goals. These goals are represented by combination of targets targets values, which are uniformly sampled by interpolating between the lower and upper bound of all targets appeared in training.

## 4.2 DIFFERENTIABLE PHYSICS GRADIENT ANALYSIS

One concern in differentiable physics based learning is the that the gradient may explode during the back-propagation through long simulation steps. Here we visualize the distribution of gradients norm of a whole training process in Fig. 6. It can be observed that most values of the gradients are distributed inside the interval [-10, 10] in log scale, which indicates that our learning approach provides stable gradients during training.

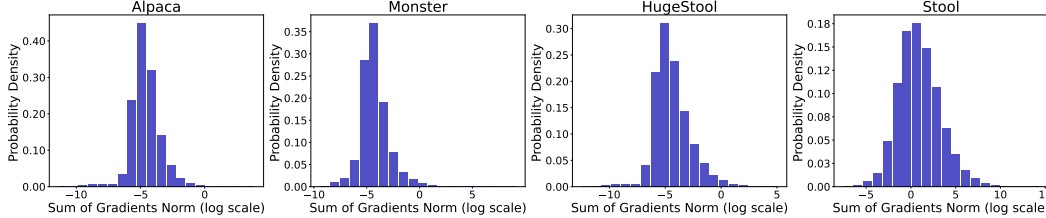

Figure 6: **Gradient Analysis.** The plots show the gradient distribution of different agent. The x-axis represent the sum of gradients norm. The values are drawn in log scale for a clear visualization. We record the sum of gradients norm of 10000 iterations for one training. For each agent, the experiments are performed at least three times, i.e., there are at least 30000 gradient samples for one agent.

**Adam vs SGD.** Hu et al. (2020) shows that stochastic gradient descent (SGD) with differentiable physics can achieve satisfying results for tasks with single goals, e.g. running along one direction. However, this no longer holds for tasks with multiple goals. To investigate the differences, we perform a learning task with multiple goals on a 2D mass spring system. In this experiment shown in the left of Fig. 5, SGD (curve Full-OP) makes minor progress while our modified Adam optimizer with reduced momentum drops the validation obviously within 10000 iterations. One key difference between Adam and SGD is that Adam utilises the momentum while vanilla SGD doesn't. To further

| $\beta_2$ \ $\beta_1$ | 0.968 | 0.9 | 0.82 | 0.68 | 0.43 |
|---|---|---|---|---|---|
| 0.999 | 0.791 | 0.799* | 0.764 | 0.783 | 0.781 |
| 0.99 | 0.763 | 0.795 | 0.786 | 0.757 | 0.756 |
| 0.968 | 0.773 | 0.776 | 0.788 | 0.772 | 0.773 |
| 0.9 | 0.799 | **0.745** | 0.760 | 0.783 | 0.817 |
| 0.68 | 0.804 | 0.756 | 0.777 | 0.767 | 0.776 |

Table 2: **Grid search for Adam hyperparameters.** The values in the table represent the normalized validation loss for *Task*. We chose 5 different values for $\beta_1$ and $\beta_2$ in Adam, which are sampled in logarithmic scale. For each setting, the experiments are repeated for multiple times. *The default values for Adam hyperparameters are $\beta_1 = 0.9$ and $\beta_2 = 0.999$.

investigate the the effectiveness of the momentum hyperparameters $\beta$, we perform a grid search for both $\beta_1$ and $\beta_2$ whose default values are 0.9 and 0.999 respectively. The results presented in table 2 show that reducing $\beta_2$ to 0.9 delivers us the best performance. We therefore use this Adam optimizer with reduced momentum in all our experiments.

### 4.3 LEARNING TECHNIQUES

**Batching** We adopt batching into differentiable physics and perform a series of experiments as shown in the right of Fig. 5 to verify its effectiveness, with batch size 1, 8, 32, and 64. It can be observed that without batching, there is a large variance in training loss and it hardly converges. Batching with a proper size helps reduce the variance and effectively improve the training performance. We also find that an overlarge batching size does not help training obviously. For example, batching with size 64 only improve the performance marginally.

**Loss design** We compare our tailored loss function with a naive loss function. The naive loss function defines the velocity as the position difference of the center of mass between consecutive time steps, i.e., requiring the agent to move at a constant velocity at each time step. The experiments show that the agent under the naive loss design can not be properly trained to run, i.e., there are almost no progress in running loss dropping.

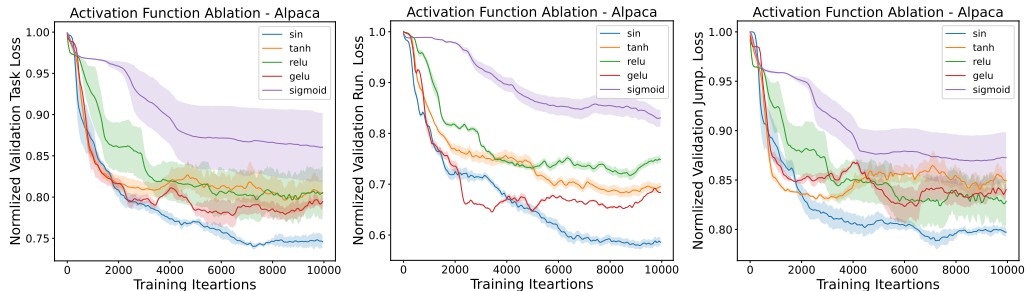

Figure 7: **Activation functions ablation study.** The figure show the validation loss between choices of different activation functions. From left to right, the subplot shows the results of *Task*, *Run* and *Jump* loss, respectively.

**Activation function** To validate the effectiveness of our $sin$ activation function, we replaced it for hidden layer with various popular activation functions $tanh$, $relu$, $gelu$ and $sigmoid$. For the output layer, the output actuation is expected to be limited in the range -1 to 1 due to the physical constraints of an actuator, e.g., a spring should not be overly compressed or stretched. Therefore we made some modifications for the activation functions in the output layer. To be more specific, we clamp the values larger than 1 or smaller than -1. In addition, for activation functions whose values are constantly zero (or very small) when input is negative, we modify the $relu$ and $gelu$ by subtracting the value by 1, for $sigmoid$ we map the its value from [0, 1] to [-1, 1]. The results in Fig.7 show that using $sin$ as activation functions gives the best performance among all three losses. For more results of activation functions ablation studies, please check the appendix

## 4.4 IMPORTANCE OF INPUT FEATURES

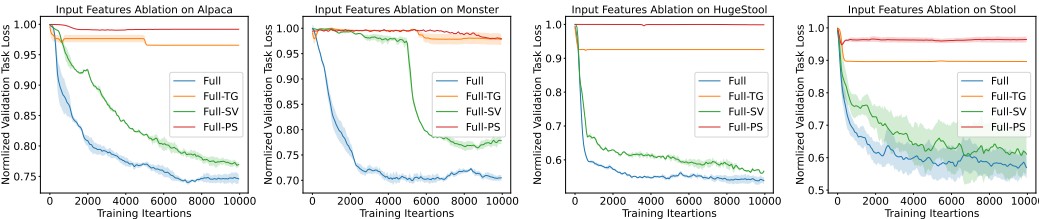

Figure 8: **Input features ablation study.** Full-PS, Full-SV and Full-TG indicate the models trained without Periodic Signal, State Vector or Targets respectively.

**Ablation on input features**   The input features consist of three parts: periodic control signal, state vector, and targets, as mentioned in section 3.3. We investigate each part of the contribution to training by ablation studies as shown in Fig.8. The results show that each of the three parts serves as a necessary component for a converged training. The periodic control signal serves as an important role. The agents can hardly move without this signal as the corresponding task loss does not drop along the training (shown by the red curves in Fig.8). The targets are another essential part, without which the optimizer can hardly work after a minor progress as shown by the yellow curves in Fig.8. Although the state vector is less important in terms of reducing the loss, the overall task performance is degraded without the state vector as shown by the green curves in Fig.8.

## 4.5 BENCHMARK AGAINST REINFORCEMENT LEARNING

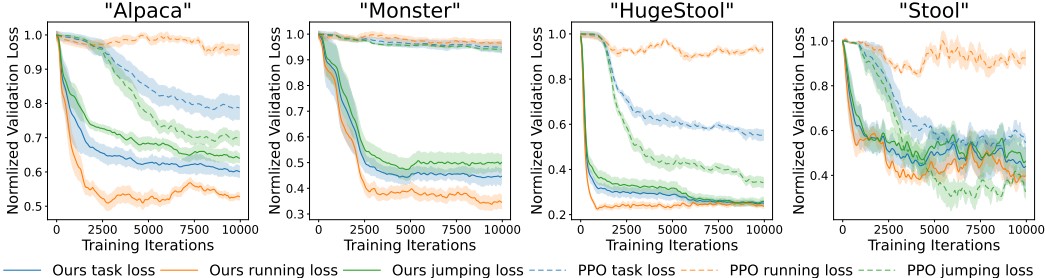

Figure 9: **Comparison on different agents.** Both our method and PPO run on GPUs. The solid and dashed lines show the validation loss of our method and PPO, respectively. The agent design is also shown in the figure. Springs marked in red are actuators.

We implement the standard proximal policy optimization (PPO) benchmark in multiple goals settings as shown in Fig. 9. The PPO agent can make progress in single goal learning and learn certain locomotion patterns to move toward the goal. Yet as the task gets complicated, PPO often gets trapped in the local minima. We try our best to fine-tune the PPO hyper-parameters, but still find it tends to overfit to one goal but fails to find a balance between different goals. It can be observed in Fig. 9 where the agents trained by PPO can learn to jump but struggle to run.

## 5 RELATED WORK

**Differentiable simulation**   Differentiable physical simulation is getting increasingly more attention in the learning community. Two families of methods exist: the first family uses neural networks to *approximate* physical simulators (Battaglia et al., 2016; Chang et al., 2016; Mrowca et al., 2018; Li et al., 2018). Differentiating the approximating NNs then yields gradients of the (approximate) physical simulation. The second family is more accurate and direct: many of these methods using differentiable programming (specifically, reverse-mode automatic differentiation) to implement physical simulators (Degrave et al., 2016; de Avila Belbute-Peres et al., 2018; Schenck & Fox, 2018;

Heiden et al., 2019; Hu et al., 2019; 2020; Huang et al., 2021). Automatic differentiation works well for explicit time integrators, but when it comes to implicit time integration, people often adopt the adjoint methods (Bern et al., 2019; Geilinger et al., 2020), LCP (de Avila Belbute-Peres et al., 2018) and QR decompositions (Liang et al., 2019; Qiao et al., 2020). In this work, we leverage Diff-Taichi (Hu et al., 2020), an automatic differentiation system, to create high-performance parallel differentiable simulators and the built-in NN controllers.

**Locomotion skill learning**  Producing physically plausible characters with various locomotion skills is a challenging problem. One classic approach is to manually design locomotion controllers subject to physics laws of motion in order to generate walking Ye & Liu (2010); Coros et al. (2010) or bicycling Tan et al. (2014) characters. These physically-based controllers often rely on a complex set of control parameters and are difficult to design and time-consuming to optimize. Another line of research suggests using data to control characters kinematically. Given a set of data clips, controllers can be made to select the best fit clip to properly react to certain situations Safonova & Hodgins (2007); Lee et al. (2010); Liu et al. (2010). These kinematic models use the motion clips to build a state machine and add transitions between similar frames in adjacent states. Although being able to produce higher quality motions than most simulation-based methods, the kinematic methods lack the ability to synthesize behaviors for unseen situations. With the recent advances in deep learning techniques, attempts have been explored to incorporate reinforcement learning (RL) into locomotion skill learning Peng et al. (2015); Liu & Hodgins (2017); Peng et al. (2018); Park et al. (2019). These modern controllers gain their effectiveness either from tracking high-quality reference motion clips or from cleverly designed rewards to imitate the reference. However, the exploration space for RL is usually prohibitively large to achieve complicated target motions. Carefully designed early termination strategies Ma et al. (2021); Won et al. (2020), better optimization methods Yang & Yin (2021), and adversarial RL schemes Peng et al. (2021) enable these RL-based methods to achieve richer behaviors. However, it is still time-consuming to get a well-trained RL model in complex tasks such as soft robot control. Our method, on the other hand, leverages the differentiable simulation framework and can achieve much better convergence behavior compared to RL-based methods.

## 6   LIMITATIONS AND CONCLUSIONS

Currently, difficulty of the learning tasks depends on robot design and physical parameters. The training performance may be degraded with improper robot designs. For instance, in our stool case where unnecessary actuators on its body are allowed, it is more difficult to achieve good results. Offloading the physical parameter tuning and robot designing to an automatic pipeline will be an interesting future research direction.

To summarize, we have presented an effective end-to-end differentiable physics based learning framework for soft robot complex locomotion skills learning. We systematically enhance classical differentiable physics learning systems with a suite of techniques and investigated the key factors contributing to the training in detail. We show that our framework can provide stable gradients without explosion during a simulation with hundreds of steps. Together with batching, the modified Adam dramatically outperforms the stochastic gradient descent (SGD) on complex tasks with multiple goals. Benefited by our tailored loss functions, the network take advantages of the three parts of the input features. We found that the periodic control signal dominates the actuation signal, while state vector and task goals have weaker effects. The over-dominated periodic signal may induce high-frequency noises. To balance the importance between the input features, we additionally pose an actuation loss as a regularization term. It can effectively suppress the noises, which help the agents move more smoothly. In addition, we compare our method with the state-of-the-art reinforcement learning proximal policy optimization. Our method shows advantages in both training performance and convergence efficiency on tasks with multiple goals.

Our framework enables users to flexibly design robots and to teach them locomotion skills. The trained agents can be manipulated to smoothly switch locomotion tasks such as running, jumping, and crawling with different speeds and orientations, interactively. To the best of our knowledge, this is the first time a learning system based on differentiable physics can deliver controllers with such robustness, flexibility, and efficiency.

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

# A    REINFORCEMENT LEARNING SETTING

**Environment.**    We use the open-source implementation of PPO (Kostrikov (2018)) in our environments. Part of important hyper-parameters are listed in the table 3.

Table 3: PPO hyper-parameters

| Parameter | Values |
|---|---|
| learning rate | $2.5e-4$ |
| entropy coef | 0.01 |
| value loss coef | 0.5 |
| number of processes | 8 |
| number of simulation steps. | 1000 |

**Reward Design.**    We design a reward function for PPO based on our loss functions. Recall the velocity part of equation 4. We define the total velocity reward as:

$$\mathcal{R}_v = \lambda_v \sum_{n \in \mathcal{T}} \sum_{t \in [\mathcal{P}_r, \mathcal{P}]} g_v(n)^2 - (\tilde{\mathbf{v}}(t) - g_v(n))^2 \qquad (6)$$

If we directly split the reward into each time steps by $t$, the rewards of first $\mathcal{P}_r$ time steps would be zero and it is too difficult for PPO to learn the policy. Therefore, we modified the reward function to tackle this issue. By equation 1, we have

$$\mathcal{R}_v = \lambda_v \sum_{n \in \mathcal{T}} \sum_{t \in [\mathcal{P}_r, \mathcal{P}]} g_v(n)^2 - \left[ \frac{1}{\mathcal{P}_r} \left( \mathbf{c}(t) - \mathbf{c}(t - \mathcal{P}_r) \right) - g_v(n) \right]^2 \qquad (7)$$

$$= \frac{\lambda_v}{\mathcal{P}_r^2} \sum_{n \in \mathcal{T}} \sum_{t \in [\mathcal{P}_r, \mathcal{P}]} [\mathcal{P}_r g_v(n)]^2 - [\mathbf{c}(t) - (\mathbf{c}(t - \mathcal{P}_r) + \mathcal{P}_r g_v(n))]^2 . \qquad (8)$$

We can define a function $f_n(t', t) = [\mathcal{P}_r g_v(n)]^2 - [\mathbf{c}(t') - (\mathbf{c}(t - \mathcal{P}_r) + \mathcal{P}_r g_v(n))]^2$ and substitute it back to equation 8. The reward can be re-written as

$$\mathcal{R}_v = \frac{\lambda_v}{\mathcal{P}_r^2} \sum_{n \in \mathcal{T}} \sum_{t \in [\mathcal{P}_r, \mathcal{P}]} f_n(t, t) \qquad (9)$$

Refer to that for any $t \in \mathcal{P}$, we have $f_n(t - \mathcal{P}_r, t) = 0$, which means

$$\mathcal{R}_v = \frac{\lambda_v}{\mathcal{P}_r^2} \sum_{n \in \mathcal{T}} \sum_{t \in [\mathcal{P}_r, \mathcal{P}]} \sum_{\Delta t \in [0, \mathcal{P}_r)} f_n(t - \Delta t, t) - f_n(t - \Delta t - 1, t) \qquad (10)$$

Let $t' = t - \Delta t$

$$\mathcal{R}_v = \frac{\lambda_v}{\mathcal{P}_r^2} \sum_{n \in \mathcal{T}} \sum_{t' \in [0, \mathcal{P}]} \sum_{\Delta t = 0}^{\min(t', \mathcal{P}_r)} f_n(t', t' + \Delta t) - f_n(t' - 1, t' + \Delta t) \qquad (11)$$

$$= \frac{\lambda_v}{\mathcal{P}_r^2} \sum_{n \in \mathcal{T}} \sum_{t' \in [0, \mathcal{P}]} \sum_{\Delta t = 0}^{\min(t', \mathcal{P}_r)} - [\mathbf{c}(t') - (\mathbf{c}(t' + \Delta t - \mathcal{P}_r) + \mathcal{P}_r g_v(n))]^2 +$$

$$[\mathbf{c}(t' - 1) - (\mathbf{c}(t' + \Delta t - \mathcal{P}_r) + \mathcal{P}_r g_v(n))]^2 \qquad (12)$$

So we can split the whole reward into each time step $t$ as:

$$\mathcal{R}_v(t) = \sum_{\Delta t = 0}^{\min(t, \mathcal{P}_r)} [\mathbf{c}(t - 1) - (\mathbf{c}(t + \Delta t - \mathcal{P}_r) + \mathcal{P}_r g_v(n))]^2 -$$

$$[\mathbf{c}(t) - (\mathbf{c}(t + \Delta t - \mathcal{P}_r) + \mathcal{P}_r g_v(n))]^2 \qquad (13)$$

Recall that the goal of a jumping task is to reach a specified height goal $g_h$, the intuition for jumping reward is to encourage the agent to improve its maximum height until reaching the height.

$$\mathcal{R}_h(t) = \lambda_h \sum_{n \in \mathcal{T}} (\min_{s \in \mathbf{S}} h(s, t + n\mathcal{P}) - g_h(n))^2 - (\tilde{h}_h(n) - g_h(n))^2 \tag{14}$$

where the $\min_{s \in \mathbf{S}} h(s, t + n\mathcal{P})$ is the agent height at current time step as mentioned in 3.1 and $\tilde{h}_h(n)$ is the max height record during one task period.

## B  NETWORK ARCHITECTURE

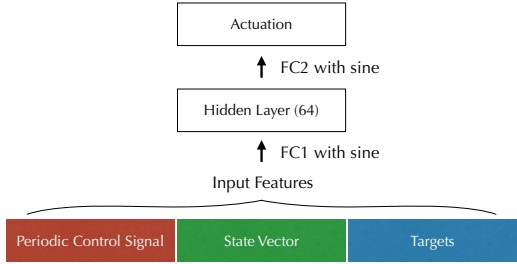

Figure 10: **Network architecture.**

Our network architecture is shown in Fig. 10. It is a two layers fully connected network with a 64 channels hidden layer. The number of input and output channels varies according to different agent designs.

## C  SIMULATION SETTING

**Mass-spring systems**  We adopt the classic Hookean spring model to represent the elastic force and use dashpot damping Baraff & Witkin (1998) as the damping force to simulate mass-spring system. We found that the drag damping model used by Hu et al. Hu et al. (2020) damps the gradient of the NN controller while the dashpot damping model is able to generate vividly changing gradients. Adopting dashpot damping model makes our agent more flexible.

**Material point methods**  We use MLS-MPM Hu et al. (2018) as our material point method simulator. We further applied the affine particle-in-cell method Jiang et al. (2015) to reduce the artificial damping. Due to the nature of MPM as a hybrid Lagrangian-Eulerian method, it always requires a background grid during the simulation. Naive MPM implementations fix the position of the background grid, hence limit the moving range of the agent. We apply a dynamic background grid that follows the agent all the time to overcome this problem.

## D  INTERACTIVE CONTROL

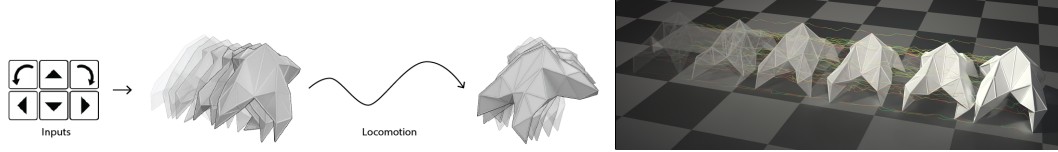

Figure 11: **Left:** Our system allows the user to control the agent's motion interactively. **Right:** the trajectory of a walking quadruped agent under user control.

Smooth interpolation between different tasks is a key advantage of our approach. In interactive settings, users can smoothly control the agent's motion via input devices such as a keyboard as shown in Fig. 11. Please refer to our supplemental video for more details.

## E  3D RESULTS

In addition to 2D cases, our framework can be applied to 3D cases seamlessly. In a 3D space, an agent is additionally expected to control its orientation on a plane, i.e., rotate. We design several 3D agents (shown in Fig. 2) and train them to run and rotate. These trained agents can achieve specified goals given control signals after trained for several hundred to a few thousand iterations. Please check the supplemental videos for more details.

**Loss function in 3D**   The agents in 3D are trained with the loss function below. The loss for running $\mathcal{L}_v$ is inherited from 2D settings. The rotation loss $\mathcal{L}_r$ is defined as the accumulation of centralized point-wise distance between current position and target rotated position.

$$\mathcal{L} = \lambda_v \underbrace{\sum_{n\in\mathcal{T}} \sum_{t=\mathcal{P}_r}^{\mathcal{P}} (\tilde{\mathbf{v}}(t) - g_v(n))^2}_{\mathcal{L}_r} + \lambda_r \underbrace{\sum_{n\in\mathcal{T}} \sum_{t=\mathcal{P}_r}^{\mathcal{P}} \sum_{s\in\mathbf{S}} ((\mathbf{s}(t) - \mathbf{c}(t)) - \mathbf{R}(\mathbf{s}(t-\mathcal{P}_r) - \mathbf{c}(t-\mathcal{P}_r)))^2}_{\mathcal{L}_r} \tag{15}$$

$$\mathbf{R} = \begin{bmatrix} \cos(\mathcal{P}_r g_\omega) & 0 & -\sin(\mathcal{P}_r g_\omega) \\ 0 & 1 & 0 \\ \sin(\mathcal{P}_r g_\omega) & 0 & \cos(\mathcal{P}_r g_\omega) \end{bmatrix}, \tag{16}$$

where $g_\omega$ is the target angular velocity assembled in input features. The format of $\mathbf{R}$ is derived by rotation matrix along XZ-plane.

Note, for solving running tasks, the rotation loss also plays an important role. A significant issue of 3D agents running is accumulated orientation error. Utilizing identity rotation matrix $\mathbf{R}$ helps adjust agent's posture. In turn's of rotation tasks, the rotation center should be maintained still which is equivalent to zero velocity. These two losses complement with each other to achieve the final goals.

**Friction model.**   In real world, agents running is driven by static friction between "feet" and ground. We configure and train agents with different contact models in our experiments. With zero friction applied, training loss shows no evidence decreasing and the agent is not able to move no matter what actions patterns it learns. With a fully sticky surface, where only upwards vertical velocity is kept and all other components of velocities are projected to zero after contact, the 2D agents work well. However, in 3D, it turns out that the agent tends to stuck. In practice, we adopt the classic slip with friction model from physically-based simulation which involves both kinetic friction and static friction. These two categories of contact are determined by how much pressure force is applied on the surface and can be treated like a branched function numerically which can be handled by our auto-differential system. Please see the supplemental video for a visual comparison between slip and sticky friction models.

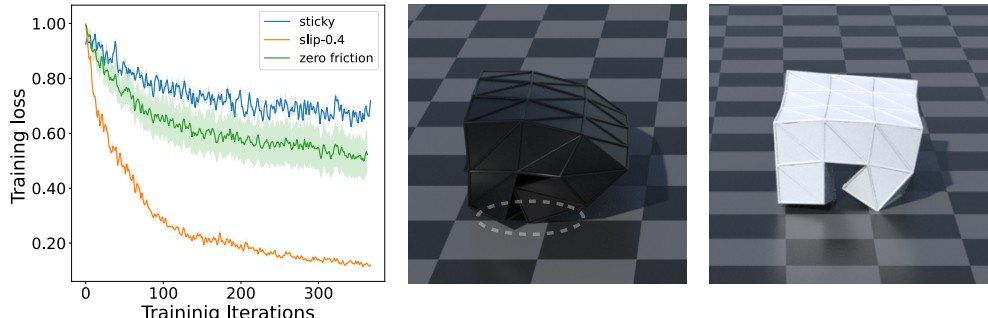

Figure 12: **Left:** Training loss for zero friction, .4 friction and sticky contact models. **Middle:** Agent suffers from sticky surface and cannot move further. **Right:** Agent moves smoothly towards right.

## F  GENERALIZATION TO MANIPULATION TASKS

Here we provide results on simple manipulation tasks in additional to locomotion tasks. In these tasks, the agent is expected to manipulate a object (marked in purple) to hit a target point (marked in green). This task has similar periodicity property like locomotion. We designed two scenarios: *Juggle* and *Dribbble and Shot* shown in Fig.13. The visual results are also shown in the video in supplemental materials.

**Juggle**   In this scenario, the target point appears in the sky above the the agent. The agent is expected to 'juggle' the object to the target point.

**Dribble and Shot**   In this scenario, the target point keeps moving toward left. The agent is expected to carry the object and shot it to the target point.

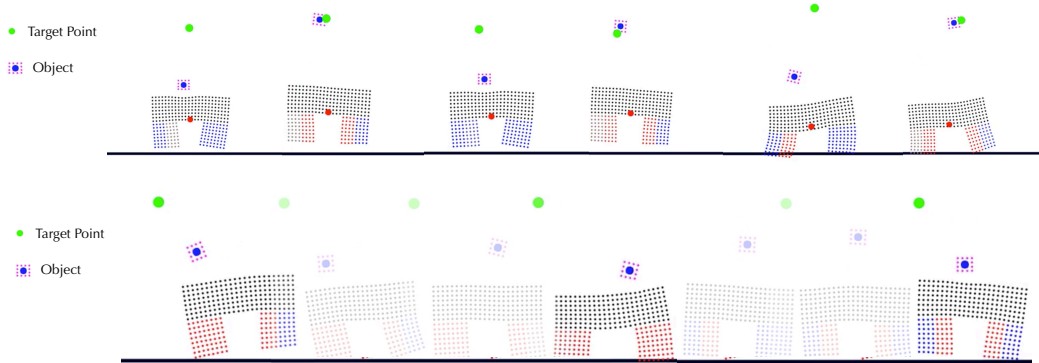

Figure 13: **Manipulation task showcase.** The upper plot shows the snapshots of scenario 'Juggle' and the lower one shows that of the 'Dribble and Shot'.

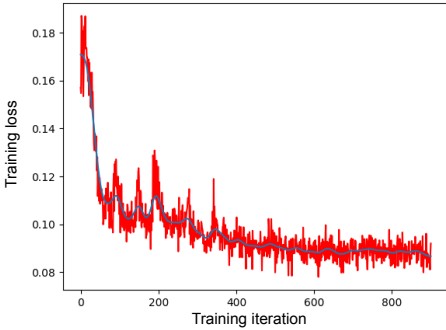

Figure 14: **Training curve of the 'Juggle' task.** The loss is defined as the distance between center of the object and the target point.

The experiments indicate that our method has the possibility to generalize to tasks beyond locomotion.

## G  DETAILED RESULTS ON FURTHER EXPERIMENTS

Here we show more detailed results on further experiments, including ablation studies on different agents, validation loss on different tasks for agents and more network weights visualization results.

Table 4: **Ablation Summary on *Monster*.**

| Name | Training Setting | | | | | | | Norm. Valid. Loss (Monster) | | |
|---|---|---|---|---|---|---|---|---|---|---|
| | OP | AF | BS | PS | SV | TG | LD | Run. | Jump. | Task |
| Full | Adam | sin | 32 | ✓ | ✓ | ✓ | ✓ | 0.46±0.01 | 0.77±0.00 | 0.71±0.01 |
| Full-OP | SGD | sin | 32 | ✓ | ✓ | ✓ | ✓ | 0.99±0.01 | 0.99±0.00 | 0.99±0.01 |
| Full-AF | Adam | tanh | 32 | ✓ | ✓ | ✓ | ✓ | 0.65±0.01 | 0.84±0.02 | 0.79±0.02 |
| Full-PS | Adam | sin | 32 | ✗ | ✓ | ✓ | ✓ | 0.93±0.00 | 0.99±0.00 | 0.98±0.00 |
| Full-SV | Adam | sin | 32 | ✓ | ✗ | ✓ | ✓ | 0.51±0.01 | 0.86±0.01 | 0.78±0.01 |
| Full-TG | Adam | sin | 32 | ✓ | ✓ | ✗ | ✓ | 0.99±0.00 | 0.97±0.01 | 0.98±0.01 |
| Full-LD | Adam | sin | 32 | ✓ | ✓ | ✓ | ✗ | 0.99±0.00 | - | 0.90±0.06 |

Table 5: **Ablation Summary on *HugeStool*.**

| Name | Training Setting | | | | | | | Norm. Valid. Loss (HugeStool) | | |
|---|---|---|---|---|---|---|---|---|---|---|
| | OP | AF | BS | PS | SV | TG | LD | Run. | Jump. | Task |
| Full | Adam | sin | 32 | ✓ | ✓ | ✓ | ✓ | 0.40±0.01 | 0.58±0.01 | 0.53±0.01 |
| Full-OP | SGD | sin | 32 | ✓ | ✓ | ✓ | ✓ | 0.97±0.01 | 0.96±0.01 | 0.96±0.01 |
| Full-AF | Adam | tanh | 32 | ✓ | ✓ | ✓ | ✓ | 0.49±0.01 | 0.59±0.01 | 0.56±0.01 |
| Full-PS | Adam | sin | 32 | ✗ | ✓ | ✓ | ✓ | 0.99±0.00 | 0.99±0.00 | 0.99±0.00 |
| Full-SV | Adam | sin | 32 | ✓ | ✗ | ✓ | ✓ | 0.40±0.01 | 0.62±0.01 | 0.57±0.00 |
| Full-TG | Adam | sin | 32 | ✓ | ✓ | ✗ | ✓ | 0.99±0.00 | 0.90±0.00 | 0.93±0.00 |
| Full-LD | Adam | sin | 32 | ✓ | ✓ | ✓ | ✗ | 0.99±0.01 | - | 0.70±0.01 |

Table 6: **Ablation Summary on *Stool*.**

| Name | Training Setting | | | | | | | Norm. Valid. Loss (Stool) | | |
|---|---|---|---|---|---|---|---|---|---|---|
| | OP | AF | BS | PS | SV | TG | LD | Run. | Jump. | Task |
| Full | Adam | sin | 32 | ✓ | ✓ | ✓ | ✓ | 0.50±0.01 | 0.58±0.04 | 0.56±0.03 |
| Full-OP | SGD | sin | 32 | ✓ | ✓ | ✓ | ✓ | 0.78±0.02 | 0.76±0.02 | 0.76±0.02 |
| Full-AF | Adam | tanh | 32 | ✓ | ✓ | ✓ | ✓ | 0.60±0.01 | 0.63±0.03 | 0.62±0.04 |
| Full-PS | Adam | sin | 32 | ✗ | ✓ | ✓ | ✓ | 0.98±0.01 | 0.96±0.00 | 0.96±0.02 |
| Full-SV | Adam | sin | 32 | ✓ | ✗ | ✓ | ✓ | 0.50±0.03 | 0.64±0.05 | 0.61±0.06 |
| Full-TG | Adam | sin | 32 | ✓ | ✓ | ✗ | ✓ | 0.98±0.00 | 0.87±0.01 | 0.89±0.01 |
| Full-LD | Adam | sin | 32 | ✓ | ✓ | ✓ | ✗ | 0.99±0.01 | - | 0.83±0.05 |

| Ouput \ Hidden | sin | tanh | relu | gelu | sigmoid |
|---|---|---|---|---|---|
| sin | **0.745** | 0.771 | 0.765 | 0.803 | 0.810 |
| tanh | 0.753 | 0.805 | 0.783 | 0.799 | 0.849 |
| relu | 0.780 | 0.770 | 0.806 | 0.775 | 0.889 |
| gelu | 0.767 | 0.780 | 0.773 | 0.795 | 0.829 |
| sigmoid | 0.754 | 0.767 | 0.800 | 0.799 | 0.860 |

Table 7: **Full ablation studies for activation functions.** The values in the table represent the normalized validation loss for *Task*. For each setting, the experiments are repeated for multiple times. It can be observed that the model using $sin$ for both hidden and output layer achieves the best results.

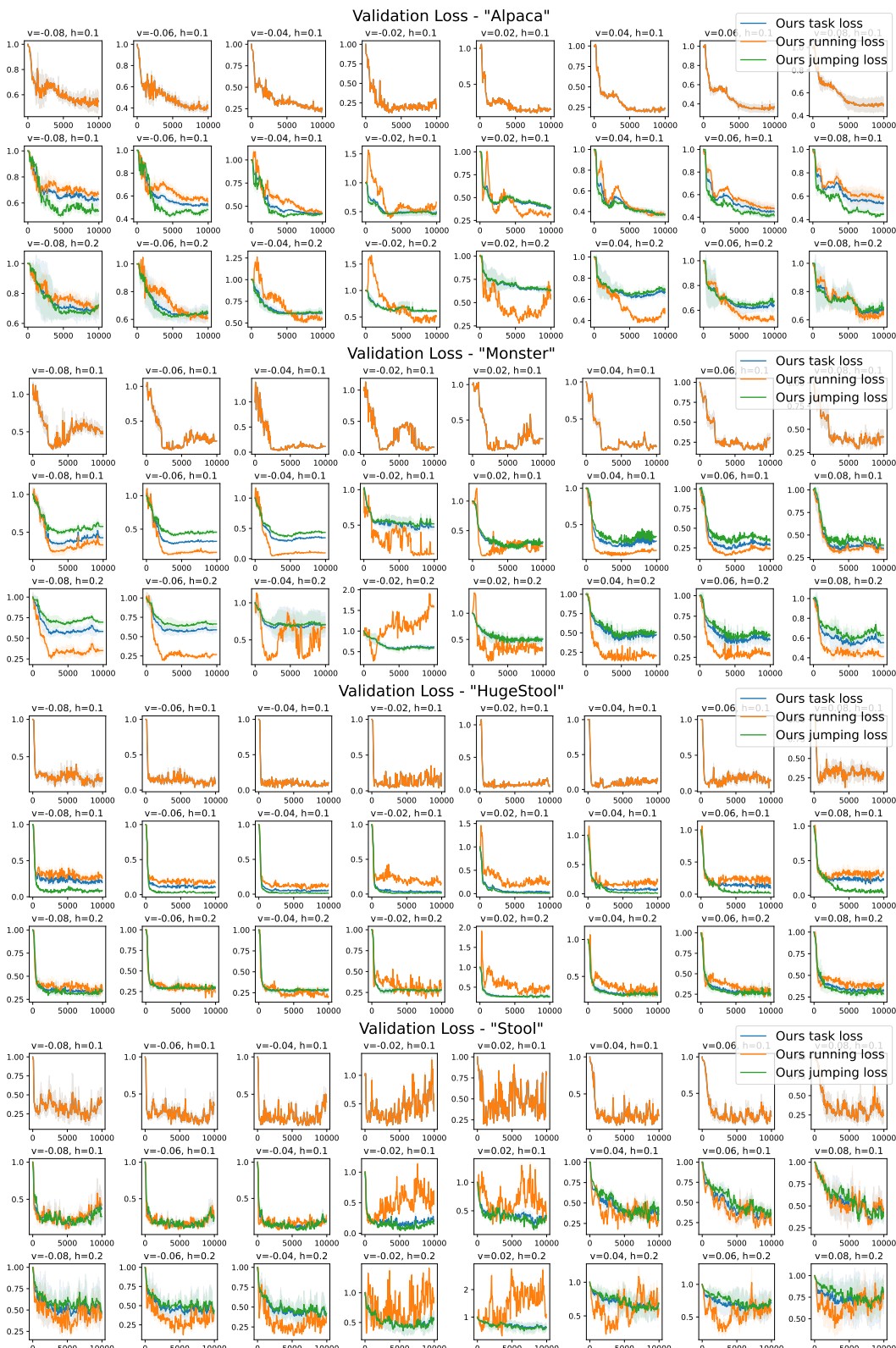

Figure 15: **Validation loss on different agents and targets.** The plot shows the validation loss on different targets combinations for agents. The target velocity ranges from -0.08 to 0.08 and the target heights are 0.1, 0.15 and 0.20. Each subplot shows the task, running and jumping loss for one agent given a pair of specified target velocity and height.

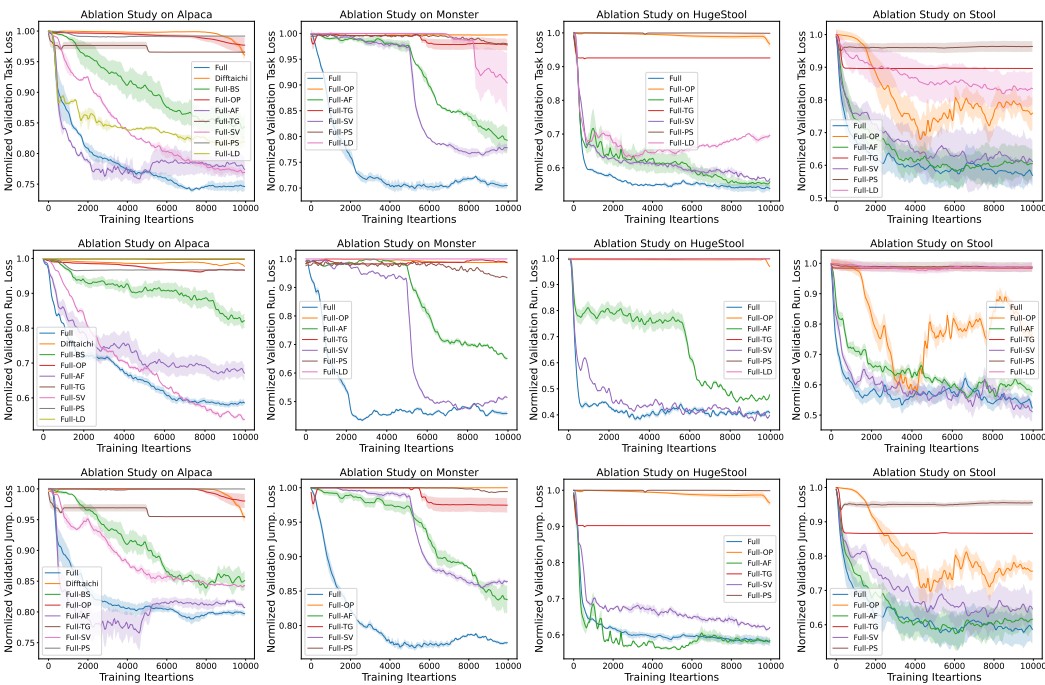

Figure 16: **Summary of the ablation study for agents.** The subplots show the normalized validation loss for task (weighted summation of different losses), running and jumping from top to bottom row. The subplots in different columns show the results of different agents.

