# OpenReview forum: "Complex Locomotion Skill Learning via Differentiable Physics"
_ICLR.cc/2022/Conference — ICLR 2022 Submitted_

### Official Review · Reviewer_cp7c · 2021-10-31

**Correctness:** 3
**Technical Novelty And Significance:** 2
**Empirical Novelty And Significance:** 3
**Recommendation:** 5
**Confidence:** 4

**Main Review:**

### Strength
* +++ The paper shows great results of interactively controlling soft robot locomotion and switching among different goals.

### Weakness
* --- The overall technical contribution is limited. The proposed contribution on loss designs is limited to the locomotion tasks, and it is not clear whether these losses would generalize to other robots and the real world [1, 2].
* -- The use of the sine activation function is not well supported by the experiments. In table 1, sine function shows marginal improvement over tanh, and in table 5 of the appendix, using tanh shows much better performance over sine. Furthermore, tanh tends to become saturated during backpropagation of a long sequence. I would suggest the author also compares with ReLU.
* -- Presentation in the paper can be improved:
     1. The modified Adam optimizer is listed as one of the technical contributions. However, it is not stated clearly what is the modification and how does it compare to the original Adam optimizer.
     2. The name of the robots are presented in Fig. 7 of Sec. 4.5 but are referenced multiple times in previous sections and table 1. It would be helpful to clarify the name of the agents early on.
     3. Reference of Fig. 5 at the end of Sec. 4.2 seems to be incorrect, as no comparison of SGD and Adam is shown in Fig. 5.

[1] Lee, Joonho, et al. "Learning quadrupedal locomotion over challenging terrain." Science robotics 5.47 (2020).
[2] Zhao, Allan, et al. "RoboGrammar: graph grammar for terrain-optimized robot design." ACM Transactions on Graphics (TOG) 39.6 (2020): 1-16.

**Summary Of The Paper:**

The paper proposes to learn neural network controllers in a differentiable simulator for locomotion tasks. A set of technical improvements are made and make the controllers more robust and require fewer samplers to learn. These include: using periodic activation functions, design of loss functions, and using large batch size. The paper shows learned locomotion skills of running, jumping, and crawling for different soft robots, which can be interactively controlled.

**Summary Of The Review:**

The paper shows interesting results of locomotion tasks, but the overall technical contribution is limited as presented. The writing of the paper could also be further improved. I am willing to raise my score if the listed concerns can be addressed.

---

### Official Review · Reviewer_kPtv · 2021-11-01

**Correctness:** 3
**Technical Novelty And Significance:** 2
**Empirical Novelty And Significance:** 3
**Recommendation:** 6
**Confidence:** 4

**Main Review:**

Strengths:
1) This paper combines differentiable simulation with many carefully chosen design decisions, which enables effective learning of soft body locomotion.
2) The analysis and ablation study provides useful and practical guidance to practitioners in the field of reinforcement learning and locomotion control.

Weakness:
1) Technical contribution is lean. The Taichi differentiable simulator is an existing work. This paper seems to use it without lots of changes. The second claimed contribution "end-to-end differentiable simulation environment, including mass-spring and material point" is not well supported. The description of simulation is high level or in Section 2. It is hard to understand the scope of this contribution.

2) I hope that the paper can demonstrate more impressive and diverse results. For example, a soft character navigating uneven terrains, or a soft robot manipulating objects. A soft body character running on flat ground, as shown in this paper, has been demonstrated in several prior works, such as DiffTaichi. Even though this paper demonstrates multiple locomotion skills (turning, different speed and jumping) in one policy, the improvement over the prior work is incremental.

Additional questions and comments:
1) The paper claims in Section 2.1 that "Any physically-based deformable object simulator can be applied to our learning framework". I thought that the simulator needs to be differentiable. So only a small subset of simulation will work. Did I miss anything?

2) It is unclear to me how applicable the results of the ablation study are to other tasks. For example, the sine actuation function and the periodic signal may not be as important for manipulation tasks. If the application of this paper is only locomotion, its contributions are quite limited.


**Summary Of The Paper:**

This paper proposes to use differentiable simulators for learning locomotion skills for deformable characters. The key contribution is the experiments and analysis of a large number of design choices, including activation function, reward design, optimizer, batch size, etc. The paper finds that by combining differentiable simulator with the right design choices, the character can learn general and complex skills with one policy. The paper also shows that this combination can significantly outperform PPO.

**Summary Of The Review:**

The paper demonstrates good results for learning soft body locomotion. Although the technical novelty is lean, I find that the analysis and ablation of various design decisions are helpful, and has practical values to guide further development in this direction. For this reason, I am slightly leaning towards accepting the paper.

---

### Official Review · Reviewer_2nYD · 2021-11-02

**Correctness:** 4
**Technical Novelty And Significance:** 2
**Empirical Novelty And Significance:** 3
**Recommendation:** 6
**Confidence:** 3

**Main Review:**

Based on existing differentiable physics engines, this paper further gives empirical recipes on how to use the physics gradients to train powerful controllers, which can help the community to develop techniques related to differentiable physics.

This method divides long-horizon tasks into smaller periods with different objectives. To avoid sparse and non-smooth rewards, objective functions are manually designed for subtasks. Periodic control input is also added to the input to actuate the controller. When training the network, the authors use batching, regularization terms, and sine activation layer to accelerate and stabilize the convergence.

When reading this paper, I have several concerns,
1. In Equation 5, how is $g_v(n)$ subtracted from $A(t)$? Do they have the same dimension? $g_v(n)$ as a velocity should have a dimension of 3 (or 6 if rotation is included). The dim of  $A(t)$ should be the DOF of actuation. Maybe you mean $||g_v(n)||$. Why only target velocity is considered, how about the height $g_h(n)$? Why “periodic control signal may induce unwanted high-frequency noises”？
2. The paper mentions "a modified Adam optimizer". I did not find how Adam is modified.
3. In Figure 4, what is the definition of one iteration? Is it one step of network update or one step of simulation?


**Summary Of The Paper:**

This paper provides a practical guide on how to train an NN-based controller for locomotion tasks with robustness and efficiency.

The tasks are categorized into motion primitives including running, jumping, and crawling, each with a manually designed objective function. Considering the periodic nature of locomotion, the author proposed to use sine activation layers and periodic signals in different phases as input. The experiments show that a larger batch size can lead to faster convergence. Compared to SGD, this paper points out that Adam is better for the tasks. The supplementary video presents that the trained networks enable real-time control of complex skills.


**Summary Of The Review:**

This paper gives us a good example of how to train a network using differentiable physics. Their video of interactively controlling 2D and 3D robots looks interesting. Although some of the techniques like staging, batching, since activation, and periodic signals are intuitive and existing tricks, it is good to see that a combination of them can successfully train a controller.

---

> ### Comment · Reviewer_2nYD · 2021-11-22
> **Thanks for your response**
>
> Dear authors,
>
> Thank you for the explanation. For the Modified Adam, I suggest it might be more accurate to just use Adam according to your description. It is common to change the default momentum parameter.
>
> You mentioned that the iteration number in Fig. 5 represents the network update. It might be unfair since larger batch sizes will see more samples. (for batch size k, you will run k different simulation).
>
> Best,
> 2nYD

---

### Official Review · Reviewer_38Wd · 2021-11-03

**Correctness:** 3
**Technical Novelty And Significance:** 2
**Empirical Novelty And Significance:** 3
**Recommendation:** 6
**Confidence:** 4

**Main Review:**

## Strengths

* The loss function design is clear and well explained, it is specifically tailored for cyclic motion which is the main task addressed by the paper.
* The use of SIREN to process the input and output a periodical signal is quite interesting. It suits well to tasks requiring cyclic motion.
* The comparison to PPO seems fair and thorough, I appreciate that the authors put efforts into having a well-tuned baseline, something which is often overlooked.
* The control of complex mass spring system is impressive

## Weaknesses

* The claim: "To the best of our knowledge, we proposed the first differentiable physics based learning frame-
work for complex locomotion tasks using single network." is a bit fishy.
Many RL algorithms handle locomotion tasks of humanoid-like robots in simulation with a single neural network. I understand the current framework use differentiable physics for mass-spring systems, and thus is not directly applicable to classical RL benchmarks without a differentiable physics simulator. Such a claim could be made if it solved classical RL locomotion tasks + harder ones for example.

* The experiment section is not really convincing and hard to follow.
** Table 1 is barely commented, there is only a bold number in one column. Only tanh activation function is compared to sin, while relu, gelu, sigmoid could be used for hidden layers with an activation function outputting a value in [-1, 1] for the final layer of the network. Given that using SIREN seems to be an important component, the paper would benefit from a better ablation study.
** The importance of features section is really unclear.
There are sentences like "The results show that each of the three parts serves as a necessary
component for a converged training. The periodic control signal serves as an important role. The
agent can hardly move without this signal" without reference to a table or a figure, so I do not know what to look to confirm the statement.
Figure 6 is also hard to read and I am not sure if the section is super insightful. To my understanding it validates the design choice of the input feature which is a bit minor compared to other components of the model.
* In the abstract, a modified Adam optimizer is mentioned "we find our adoption of batching and modified Adam optimizer effective in training complex locomotion tasks". It seems to be important according to the authors but there is no experiment on vanilla Adam vs the proposed one with reduced momentum so we cannot asses how useful it is.

## Other comments

* The abbreviations of Table 1 could be put as caption rather than footnote for readability. The back-and-forth between table and footnote is not great for the reader.


**Summary Of The Paper:**

This work proposes to learn locomotion skills on soft robots made of springs. It proposes to leverage differentiable physics along with a NN controller based on SIREN, which allows to directly learn policies by minimizing loss functions defined on trajectories. The methods allows to learn locomotion and jump behaviors on a various set of 2D and 3D mass-spring systems.

**Summary Of The Review:**

The paper is interesting, proposing a clean framework to perform locomotion tasks on mass-spring systems. The benefit of using differentiable physics is clearly highlighted and the use of SIREN to learn periodic motions is on point. However the experimental section is weak. For this reason I only grade it as weak-reject but would be inclined to bump it if this part is strengthened.

---

### Decision · Program_Chairs · 2022-01-20

**Decision:**

Reject

**Comment:**

A nice paper and very close to being good.  But the focus on hyperparameter tuning of the optimisation method is really not novel, and the experimental validation is not strong enough.  With both theory and experimental just being marginal improvements, the paper is not considered quite ready yet.  Strong suggestion to improve on the weaknesses of the paper and resubmit – next time you'll have a clear acceptance.